# Using Electronic Medical Records to Identify Potentially Eligible Study Subjects for Lung Cancer Screening with Biomarkers

**DOI:** 10.3390/cancers13215449

**Published:** 2021-10-29

**Authors:** Lamorna Brown, Utkarsh Agrawal, Frank Sullivan

**Affiliations:** School of Medicine, University of St Andrews, St Andrews KY16 9AJ, UK; ua1@st-andrews.ac.uk (U.A.); fms20@st-andrews.ac.uk (F.S.)

**Keywords:** cancer, screening, smoking, electronic records

## Abstract

**Simple Summary:**

Recent cancer screening trials have found that using low-dose computed tomography (LDCT), compared to chest radiography, resulted in a significant reduction in lung cancer mortality. To effectively carry out this intervention, individuals at a high risk of developing lung cancer are targeted. However, accurately identifying and retaining these groups can be challenging. As electronic medical records (EMRs) contain important demographic and clinical information, they could be used to accurately identify subjects for screening. To determine whether EMRs can be used for this purpose, this paper examines the evidence around the use of EMRs in screening trials and the information contained in them that could be used to aid researchers in identifying eligible subjects.

**Abstract:**

Lung cancer screening trials using low-dose computed tomography (LDCT) show reduced late-stage diagnosis and mortality rates. These trials have identified high-risk groups that would benefit from screening. However, these sub-populations can be difficult to access and retain in trials. Implementation of national screening programmes further suggests that there is poor uptake in eligible populations. A new approach to participant selection may be more effective. Electronic medical records (EMRs) are a viable alternative to population-based or health registries, as they contain detailed clinical and demographic information. Trials have identified that e-screening using EMRs has improved trial retention and eligible subject identification. As such, this paper argues for greater use of EMRs in trial recruitment and screening programmes. Moreover, this opinion paper explores the current issues in and approaches to lung cancer screening, whether records can be used to identify eligible subjects for screening and the challenges that researchers face when using EMR data.

## 1. Introduction

Lung cancer remains one of the most aggressive and frequently diagnosed cancers in the UK [1]. Mortality rates for the disease remain high, at 21% for both males and females, making it the most common cause of cancer-related death [2]. As late-stage lung cancer (i.e., stage III/IV) is less susceptible to curative medical interventions, such as surgical resection, there is a low survival rate for individuals diagnosed at these stages (2–3%) [2]. The majority of lung cancer cases are diagnosed with late-stage cancer, leading to overall low survival rates at 1 (40%) and 5 years (16%) post-diagnosis [1,3,4].

To reduce late-stage diagnosis, lung cancer screening using low-dose computed tomography (LDCT) has been recommended [5]. Screening trials using LDCT, compared to usual care (i.e., chest X-rays), have provided evidence of a significant mortality benefit. Trials such as the NLST, NELSON and UK Lung Cancer Screening Trial found those undergoing LDCT scans had a reduced probability of dying from lung cancer [6,7,8]. The Early Detection of Cancer of the Lung Scotland (ECLS) trial also indicated that blood-based biomarkers are effective when used in conjunction with LDCT, significantly reducing late-stage diagnosis and lung cancer mortality [9].

While these trials support the use of LDCT in screening programmes to identify lung cancer, there are practical barriers that can reduce participant engagement, limiting the effectiveness of interventions [10]. These practical barriers include difficulties accessing target groups and identifying patients that fit screening inclusion criteria [10,11,12]. However, electronic medical records (EMRs) contain important clinical and demographic information that can reduce and resolve these issues [13].

This paper covers the current issues in and approaches to lung cancer screening and appraises the methods used and evidence for the effectiveness and appropriateness of using electronic medical records as a way of identifying those at high risk of developing cancer.

Defining high-risk groups for lung cancer screening is an ongoing challenge. Age, occupation, family history, some respiratory conditions (particularly emphysema) and environmental factors such as air pollution and radon exposure are important risk factors for lung cancer [14,15]. The strongest determinant of lung cancer, however, is smoking, with over 70% of cases in the UK linked to smoking [16,17]. As a result, smoking status has been used to identify eligible participants for lung cancer trials. In this article, we consider an important characteristic of high-risk groups to be whether they are current smokers, and thus papers which report on the recording of smoking in EMRs in order to identify eligible subjects are included in this article. Other health, sociodemographic and environmental risk factors for lung cancer that appear in EMRs are also examined.

## 2. Issues and Approaches to Current Lung Cancer Screening Programmes

Lung cancer screening programmes use a targeted approach, whereby those most at risk, and thus most likely to benefit from screening, are eligible for inclusion. Trials such as the NELSON and NLST use patient self-declared age and the number of pack years as bases for inclusion using a questionnaire [6,18]. Trials utilising risk models to identify high-risk groups have provided further risk factors to consider for screening criteria, such as family history of lung cancer and respiratory diseases [19]. The use of these models for participant selection has led to lower numbers of individuals eligible for selection but enabled greater prevention of lung cancer death in trials [15,20].

Despite progress in the identification of high-risk individuals, low participation and retention rates can hinder the effectiveness of interventions. Table 1 presents the approach response rates, methods of recruitment and percentage of respondents randomised for some of the major European lung cancer screening trials. Previous lung cancer trials have had approach response rates (i.e., the proportion of individuals who responded when approached) between 23 and 52% [21]. To improve these rates, the barriers and issues around lung cancer screening implementation must be explored.

There are both participant- and provider-related barriers to lung cancer screening engagement. The UK Lung Cancer Screening Pilot Trial identified participant demographic factors associated with a reduced likelihood of participation. It was found that those who were female, older, current smokers and from a lower socioeconomic group were less likely to participate [27]. Further, there are both emotional and practical barriers to participation. Practical barriers such as a participant’s state of health and emotional barriers such as fear of screening and information avoidance are cited as reasons for non-participation by eligible subjects [27,28,29]. The stigma associated with lung cancer may also act as a barrier for both participants and providers [30,31]. Patients with lung cancer report feeling more stigmatised by themselves and others compared to individuals with cancers such as breast, cervical and skin cancer, as there is a perception that they have brought the illness upon themselves by smoking [32]. This can delay individuals seeking help and receiving timely investigation and treatment, which can have a detrimental effect on patient outcomes [33]. Stigma is also associated with reduced levels of screening uptake [34].

The significant barriers that providers face relate to identifying and recruiting eligible subjects. Previous lung cancer screening trials identified subjects through population-based registries [21]. Information that could aid in the identification of high-risk groups may not be present in these registries. Additionally, the information that is present may not be accurate and, as a result, researchers risk contacting individuals who do not meet trial eligibility criteria. Trials that use electronic medical records (EMRs) for identifying subjects have shown that both identification and uptake can match those of trials that have utilised population registries. The LHC Liverpool study utilised EMRs to search for eligible subjects before contacting them; this targeted approach resulted in the trial obtaining one of the highest approach response proportions out of recent lung cancer screening trials (40%) [21,26]. The ECLS trial similarly searched for eligible participants through primary care EMRs. This trial recruited 12,208 participants and is, consequently, the largest trial for the detection of lung cancer using blood-based biomarkers [9,35]. Additionally, the ECLS and both LHC trials had a lower percentage of respondents drop out between response to invitation and randomisation (see Table 1). This indicates that EMRs can potentially aid researchers in identifying and retaining eligible study subjects.

## 3. Can Records Be Used to Aid in Identifying Eligible Subjects for Screening?

EMRs have been used to aid in identifying patients eligible for screening. A large-scale study in Minhang District in China, conducted between 2008 and 2016, used EMRs of 5 million patients to identify those eligible for screening multiple cancers including colorectal, gastric, liver, lung, cervical and breast cancer [36]. As a result, more cases of cancer were detected at an early stage, including a number of individuals who were identified as being at high risk of cancer. Similarly, trials for Lung Health Check programmes, implemented in Liverpool and Manchester, were able to recruit and retain a significant proportion of respondents approached [9,26]. These studies indicate that EMRs could be used to conduct more focused interventions. In addition, previous studies have also used machine learning algorithms on smoking history information, identified from EMRs, to create a registry of patients eligible for cancer control efforts, such as smoking cessation and lung cancer screening, which could additionally aid in targeting eligible patients for screening [37,38].

### 3.1. What Codes Are Associated with LC and Appear in EMRs?

Codes are frequently used to identify patients with various health conditions. Published comorbidity indices and phenotype code lists, such as CALIBER, the Charlson Comorbidity Index, the Elixhauser Comorbidity Index and the Quality and Outcomes Framework (QOF), have compiled a list of codes for lung cancer [39,40,41,42,43]. Moreover, different coding formats are used within different data sources in the EMRs, for example, primary care settings use read codes and secondary care settings use ICD codes [44,45]. A sample code list is presented in Appendix A, Table A1.

Various smoking codes are present within EMRs. These can be used to identify high-risk smokers for screening. Wiley et al. (2013) and Atkinson et al. (2018) examined whether smoking read codes present in EMRs could be used to determine the smoking status of participants [46,47]. Wiley et al. used ICD-9 smoking codes and found that they could accurately detect true smokers in a general population [46]. The combination of codes and free text improved sensitivity to ever smokers, however. Atkinson et al. used smoking read codes found in primary care general practice records to assess participants’ smoking history [47]. They found that read codes compared well with a population health survey (Kappa–0.64), indicating that read codes are moderately accurate and, thus, can be used in the identification of smokers.

Codes for health conditions and environmental factors present in EMRs could also be used to identify high-risk groups. A study utilising EMRs from general practices across the UK found that asbestos exposure, COPD and symptoms such as coughing and chest pain were frequently recorded in EMR documentation and prevalent among those diagnosed with lung cancer [48]. Further to this, COPD recording has been explored in EMRs. Algorithms have been developed to determine the presence of COPD in patients. Quint et al. (2014) and Chu et al. (2021) developed two such algorithms that performed well, with positive predictive values (PPVs) of 86.5% and 93.5% [49,50].

Other risk factors such as alcohol consumption and asthma have also been examined. Read codes for alcohol consumption have been validated by comparing EMR data to a health survey. The study by Mansfield et al. (2019) found similar prevalence rates between both a health survey and an EMR dataset, indicating EMRs can be accurately used to identify both current and non-drinkers [51]. Asthma has been validated in EMRs, with the PPVs of studies comparing asthma codes to a reference ranging from 46 to 100% [52].

While there are other social and environmental determinants of lung cancer, such as air pollution and radon exposure, this detailed information is not routinely collected in EMRs. To examine environmental factors, recent studies have linked geospatial and environmental data to EMRs in order to examine related health outcomes [53,54,55]. Greater consensus on measures to be captured in EMRs, as well as improvements in the linking of external sources of environmental data, could address this issue.

### 3.2. Use of Free Text to Identify Eligible Participants?

Most studies have used structured variables such as smoking status (non-smoker; ex-smoker; light smoker; moderate smoker; heavy smoker), asthma diagnosed ever (yes/no), pneumonia diagnosed ever (yes/no) and family history of lung cancer (yes/no) to estimate the risk of having lung cancer and to identify participants eligible for lung cancer screening studies [19,56,57]. However, recent studies have begun to explore free text in EMRs to identify eligible patients [58,59,60].

Natural language processing provides a feasible way to extract various types of information from EMRs. This technique has been successfully used to extract and quantify smoking information in EMRs. De Silva et al. and Palmer et al. used text analysis to quantify pack years from EMR free text [61,62]. This was successfully performed for the majority of cases, but due to the heterogeneity of clinical notes, mis-categorisation and missing cases remained an issue. Smoking status can also be identified accurately from EMRs. Groenhof et al. extracted information on smoking behaviours from free text to categorise participants into current, past and never smokers. Smoking information was accurately retrieved for the majority of cases [63].

This method of smoker identification may be more accurate and less costly and time consuming compared to asking potential participants to fill out questionnaires or to assess their own eligibility for screening. Indeed, free text in EMRs has provided more accurate and comprehensive information on smoking than structured sources of data from EMRs [64]. As these papers indicate that smoking information is present in EMRs and that smokers and non-smokers can be accurately identified from the information contained in them, this method of identification may be feasible for participant identification.

## 4. What Are the Challenges in Using EMR Data to Detect and Identify High-Risk Populations?

While, when utilising EMR data, screening programmes may achieve better targeting of eligible subjects, there are significant challenges to using EMR data. Data completeness for certain coded data elements can vary, with diagnostic and lifestyle data being less populated than prescription data [62]. Indeed, two prevalent issues affecting data completeness are missing data elements and errors in the recording of health conditions/lifestyle factors. Martin found 43% of the electronic records examined contained errors. Indeed, multiple errors were found in participant records which resulted in a total of 229 errors in 169 participant records [65]. Marston et al.’s study found that 20% of their sample had missing smoking data [66]. While overall trends show that the recording of risk factors such as smoking status has improved, missing data are still a concern, with recorded information on health care indicators only present in 10–40% of sampled EMRs [67,68,69,70].

The accuracy and quality of EMR data are a further issue. This is usually examined by comparing coded or extracted EMR data against a “gold standard” reference. Studies examining data quality show mixed results. Booth et al. examined CPRD data compared to population survey data [71]. They found little difference between the prevalence of smoking in CPRD data compared to the population survey. Estimates for current smokers and non-smokers were similar to survey data estimates, but there was underreporting of former smokers in EMRs. Similarly, asthma recording in EMRs was found to compare moderately well with manual chart reviews, with NLP and diagnosis code-based algorithms generating PPVs of 88.0% and 57.1% [72]. Conversely, Modin et al. found significant discordance between pack years recorded in EMRs and pack years determined from a shared decision-making conversation [73]. This research highlights the difficulties in truly determining data accuracy as references may not contain accurate information.

Obtaining ethical approval to access EMRs is equally challenging. EMRs contain sensitive information which means it is imperative that the data are stored and accessed in a secure way. As a result, it can be both costly and time consuming to access and obtain EMR data. Given that the use of EMR data in clinical research has grown, the development and usage of Data Safe Havens to store EMR data have mitigated some of the ethical concerns around the accessibility and storage of the data.

## 5. Future Research

There has been significant research on the extraction and classification of smoking status in EMRs. However, further research on the use of EMR information to identify and flag patients for follow-ups or screening is required. Safety netting is viewed as a best practice for those at risk of cancer, although there is little evidence for its effectiveness for cancer detection [74]. The use of EMRs to detect and flag patients for follow-ups has been successfully implemented to detect risk of adverse events, delays in follow-ups to abnormal lung imagining findings and delays in cancer diagnosis [75,76,77]. Algorithms that detect delays in follow-ups have identified a lack of appropriate follow-up action based on four diagnostic cues. The same could be performed to investigate their use for flagging patients that either partially or fully meet screening criteria.

While there is a significant amount of research examining the validity of smoking behaviours in EMRs, further research could be conducted to examine quality for other data elements. There are few papers examining environmental factors such as asbestos and radon exposure. Examining the completeness, accuracy and frequency of recordings for these exposures could aid in identifying high-risk populations.

Further research on lung cancer risk modelling using EMR data is also required [6]. Many risk models have been developed which include clinical and demographic factors. These models utilise trial or registry data and, as a result, there is a lack of research examining the use of real-world EMR information and the use of linked datasets in risk modelling [78]. Wang et al. used EMR data to model the incidence of lung cancer, and they were able to extract a large number of features to include, demonstrating the usefulness of EMR data in modelling [79]. Additionally, further examination of risk models using EMR data would be useful to identify whether models apply well to other datasets.

## 6. Conclusions

Lung cancer screening using LDCT and biomarkers has the potential to reduce late diagnosis, thereby lowering mortality rates and improving survival of the disease. However, there are significant issues with the detection of subjects eligible for lung cancer screening. Screening trials and programmes have low approach response rates, despite targeting those at a higher risk of developing cancer.

EMRs have provided useful information for clinicians and researchers which has resulted in greater engagement. For example, both the LSUT study and ECLS trial recruited a large number of participants by identifying eligible patients through EMRs. Further, the research presented in this article has shown there are data features contained in EMRs that have the ability to aid screening, such as smoking information contained in codes and free clinical text. This information can ensure that eligible populations are easier to access for researchers/clinicians and that, as a result, these individuals can be better targeted.

There are significant challenges to using EMR data such as a lack of data completeness and data accuracy. With the advances in text analysis and improvements in EMR structure and codes, they may be a viable option that both health systems and researchers can use to identify populations for lung cancer screening.

## Figures and Tables

**Table 1 cancers-13-05449-t001:** The recruitment strategies, numbers of subjects approached, numbers of respondents and the percentage of respondents randomised in major European lung cancer screening trials.

Lung Cancer Screening Trial	Recruitment Period	Number of Subjects Approached	Number of Subjects That Responded	Approach Response Rate	Number of Eligible Subjects That Consented	% of Respondents Randomised	Method of Recruitment
NELSON [6]	2003–2006	606,409	150,920	24.9%	15,822	10.5%	Direct mail
ITALUNG [22]	2004–2006	71,232	17,055	23.9%	3206	18.8%	Direct mail
LUSI [23]	2007–2011	292,440	95,797	32.8%	4052	4.2%	Direct mail and mass media
NLST [18]	2002–2004	n/a	53,454	n/a	52,486	n/a	Direct mail, mass media and outreach
UKLS [8]	2011–2014	247,354	98,746	39.9%	4061	4.1%	Direct mail
LSUT [24]	2015–2017	2012	1058	52.6%	770	72.8%	Direct mail
LHC Manchester [25]	2016–2018	16,402	2827	17.2%	1384	49.0%	Searched GP records to send direct mail invitations
LHC Liverpool [26]	2016–2018	11,526	4566	39.6%	1318	28.9%	Searched GP records to send direct mail invitations
ECLS [9]	2013–2016	77,077	18,657	24.2%	12,209	65.4%	Searched GP records to send direct mail invitations, mass media and outreach

## Data Availability

Not applicable.

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
