# Peer review of "Using Electronic Medical Records to Identify Potentially Eligible Study Subjects for Lung Cancer Screening with Biomarkers"

_cancers, 2021, doi:10.3390/cancers13215449_

Round 1
Reviewer 1 Report
This opinion paper concisely summarizes some of the challenges of recruiting patients to lung cancer screening trials. A review of relevant literature is presented and is not missing any key points. The authors might consider specifically highlighting lung cancer stigma as a patient and provider level barrier (although I'm not certain this is applicable outside the United States). For example, see https://www.ncbi.nlm.nih.gov/pmc/articles/PMC6417494/
Author Response
Reviewer #1
- The authors might consider specifically highlighting lung cancer stigma as a patient and provider level barrier (although I'm not certain this is applicable outside the United States).
Response: We agree. Lung cancer stigma has been included as a patient and provider level barrier in the 2nd section on page 3, which states:
“The stigma associated with lung cancer may also act as a barrier for both participants and providers [30, 31]. Patients with lung cancer report feeling more stigmatized by themselves and others compared to individuals with cancers such as breast, cervical and skin as there is a perception that they have brought the illness upon themselves by smoking [32]. This can delay individuals seeking help and receiving timely investigation and treatment which can have a detrimental effect on patient outcomes [33]. Stigma is also associated with lack of screening uptake [34].”
Reviewer 2 Report
The manuscript focuses on the Electronic Medical Records (EMRs) containing important demographic and clinical information, to identify subjects for lung cancer screening. The topic is very interesting and the manuscript well written, however, there are some major points to be addressed.
It is not clear how the authors identify high-risk groups have provided further risk factors to consider for screening criteria. It will be interesting to consider other exposures such as radon,passive smoke; and car traffic. Because the contribution of environmental factors in human lung cancer occurring in non-smokers. The biological function of miRNAs identified as lung cancer contributors in environemtnal signatures reflect the pivotal role of the damage to the microRNA machinery during the carcinogenesis process.
Author Response
- It is not clear how the authors identify high-risk groups have provided further risk factors to consider for screening criteria.
Response: We consider smoking to be an important risk factor and characteristic of individuals at high-risk of developing lung cancer but we have amended the introduction to include more risk factors such as occupation, air pollution and radon exposure. We have now expanded on our definition of high-risk groups and tried to make it clearer in the first section on page 2, which reads as:
“Defining high-risk groups for lung cancer screening is an ongoing challenge. Age, occupation, family history, some respiratory conditions (particularly emphysema) and environmental factors such as air pollution and radon exposure are important risk factors for lung cancer [14, 15]. The strongest determinant of lung cancer, however, is smoking, with over 70% of cases in the UK linked to smoking [16, 17]. As a result, smoking status has been used to identify eligible participants for lung cancer trials. In this article, we consider an important characteristic of high-risk groups to be whether they are current smokers and so papers reporting on recording of smoking in EMRs to identify eligible subjects will be included in this article. Other health, sociodemographic and environmental risk factors for lung cancer that appear in EMRs, will also be examined.”
- It will be interesting to consider other exposures such as radon,passive smoke; and car traffic. Because the contribution of environmental factors in human lung cancer occurring in non-smokers. The biological function of miRNAs identified as lung cancer contributors in environemtnal signatures reflect the pivotal role of the damage to the microRNA machinery during the carcinogenesis process.
Response: We have amended sections 3.1, 4 and 5 so that other exposures are considered. Asthma, COPD, asbestos, and alcohol consumption were examined. We mention that while environmental factors such as radon exposure and air pollution are associated with greater risk of incidence of lung cancer, we found that these are not routinely collected in EMRs, instead many studies link geospatial and environmental factors to EMRs to examine these variables. So, often these variables cannot directly be used to identify high-risk groups from EMRs. This was mentioned in section 3.1 on page 4.
“Codes for health conditions and environmental factors present in EMRs could also be used to identify high-risk groups. A study utilising EMRs from General Practices across the UK found that asbestos exposure, COPD, and symptoms such as coughing, and chest pain were recorded in EMR documentation and prevalent among those diagnosed with lung cancer [48]. Further to this, COPD recording has been explored in EMRs. Algorithms have been developed to determine the presence of COPD in patients. Quint et al (2014) and Chu et al (2021) developed two such algorithms that performed well, with positive predictive values (PPV) of 86.5% and 93.5% [49, 50]. Read codes for alcohol consumption have been validated by comparing EMR data to a health survey, The study by Mansfield et al (2019) found similar prevalence rates between both a health survey and an EMR dataset, indicating EMRs can be accurately used to identify both current and non-drinkers [51]. Asthma, another risk factor for lung cancer, has been validated in EMRs, with the PPVs of studies comparing asthma codes to a reference ranging from 46-100% [52].
While there are other social and environmental determinants of lung cancer, such as air pollution and radon exposure, this detailed information is not routinely collected in EMRs. To examine environmental factors, recent studies link geospatial and environmental data to EMRs in order to examine related health outcomes [53, 54, 55]. Greater consensus on measures to be captured in EMRs, as well as improvements in the linking of external sources of environmental data could address this issue.”